# A 2 MS/s Full Bandwidth Hall System with Low Offset Enabled by Randomized Spinning

**DOI:** 10.3390/s22166069

**Published:** 2022-08-14

**Authors:** Robbe Riem, Johan Raman, Pieter Rombouts

**Affiliations:** Department of Electronics and Information Systems, Ghent University, 9000 Ghent, Belgium

**Keywords:** Hall plate, current spinning, ILSA, randomized spinning, offset, spread-spectrum offset reduction loop (SS-ORL)

## Abstract

In this paper, a Hall plate readout with a randomized four-phase spinning-current scheme is proposed. The goal is to remove the maximum number of offset components, including the offset associated with spike demodulation. The outcome is that only the smallest possible offset remains, corresponding to the residual offset of the Hall plate which cannot be distinguished from the Hall signal. An additional innovation is to operate various offset-reduction loops in spread-spectrum mode, allowing the removal of error components without notching out any in-band signals. The resulting approach delivers a very large notch-free bandwidth while simultaneously reducing the Hall plate residual offset, making it an enabler for high-bandwidth Hall-based current sensors. To demonstrate the proposed techniques, we have realized a mixed-mode experimental circuit, where the analog part is implemented in a custom integrated circuit, and the digital control system in an FPGA is connected to the analog chip. Measurement results feature a Hall readout system with a notch-free bandwidth up to 820 kHz and a 47 μTrms noise floor. The input-referred Hall plate offset, based on statistical measurements on 10 samples from a single wafer, is reduced from 130±22 μT to only 23±22 μT.

## 1. Introduction

Hall plates are widely used as magnetic field sensors because they can be easily co-integrated with on-chip readout circuitry in nearly all standard Complementary Metal Oxide Semiconductor (CMOS) technologies. They form a cheap solution for various applications requiring contactless position sensing or galvanic isolated sensing of large electric currents [1,2,3,4,5,6,7]. A well-known disadvantage of Hall plates is their relatively large offsets compared to the weak information signal that is generated by the Hall effect [1,8,9]. Furthermore, this offset is sensitive to temperature and mechanical stress, forming a significant and largely unpredictable disruptive signal in the sensor measurement system.

A popular technique to combat the Hall plate offset is current spinning [10]. Here, the path of the bias current is constantly changed from one readout to the next. On a system level, the net result of current spinning is that the offset and the useful signal are separated in frequency in a way that is completely analogous to chopper modulation [11]. However, while current spinning is a key enabler for dealing with the Hall plate offset, it is not a solution on its own. The offset component is still fully present in the Hall plate output signal. Suppression of the offset-related components is in practice performed by means of filtering [11] or an offset reduction loop (ORL) [1,2,12,13,14,15,16,17,18]. A drawback of such an approach is that, in principle, it prevents the signal bandwidth from extending beyond the chopping frequency. Indeed, for a signal frequency exactly equal to the chop frequency, the spinning scheme fails to put offset and signal components at different frequencies, and therefore the signal is interpreted by the readout chain as an offset that is to be removed. In many applications, the potential removal of an in-band signal is not acceptable, which implies that the sensor bandwidth is fundamentally limited by the chop frequency resulting from current spinning. This limitation of the spinning Hall readout can be circumvented by combining the readout from multiple paths [1,2], but this comes at a cost in terms of system complexity and area.

Recently, a readout system has been presented in the literature that comes close to the maximum achievable bandwidth [13]. However, there is still an offset-related tone present at half the chop frequency, which is a known consequence of using four-phase spinning of the Hall plates [1,13]. Similar to the main chop tone, the half-rate chop tone can also be removed by means of linear filtering or an ORL. For instance, in [1,2] up to three distinct analog offset reduction loops are applied that together tackle the main and the half-rate chopper tones. Next to the cost in terms of die area, power consumption and design complexity, there is again a more fundamental problem associated with these approaches: an input signal at the half chop-rate frequency is removed as well. As a result, the (guaranteed) bandwidth of the system is in practice halved. For many applications, this may not pose a problem, e.g., for position sensing [19]. However, in current-sensing applications, there is a much higher demand for bandwidth [20], and a 50% loss in bandwidth purely because of one parasitic tone is an unfortunate side effect. Two-phase spinning [10,21] is also an option, in which case there is no half-rate tone. However, this increases the residual Hall plate offset with a significant factor (experimental results in Section 5 show a factor of over 6×). When wanting both the full bandwidth and the best possible offset reduction, the only currently known option that allows this is to apply four-phase spinning and subsequent removal of the parasitic tone by means of calibration. However, due to the finite accuracy of the calibration method, it is expected that some residual tone will be present at the half-chop frequency. So, while calibration is a possibility, there is a genuine interest in alternative approaches which are easier and economically more viable.

The purpose of the present paper is three-fold. First, we want to demonstrate that by adding randomization to the spinning scheme, in line with what is proposed in [22], all offset-related errors can be spread over the complete bandwidth, except for the fundamental residual offset of the Hall plate. This is in contrast to normal four-phase spinning where the energy of the half-rate chop tone is fully concentrated at a single frequency. The randomized approach also allows us to distinguish between many more offset components, e.g., also those caused by spike demodulation. Secondly, we show that the randomization is an enabler for specialized offset reduction loops that operate in spread-spectrum mode. Not only does this allow the removal of the different offset components, the offset reduction loops can also operate “under the radar” without notching out any in-band signals. This way, it is possible to deliver full-bandwidth readout without the need for calibration-based removal of the half-rate chopper tone. Furthermore, we present a fully digital implementation of the spread-spectrum offset reduction loops, which provides a competitive and highly flexible solution compared to prior-art analog-oriented offset reduction loops. We will show that combining the aforementioned techniques enables us to increase the spin frequency while reducing the residual DC offset. This circumvents the trend observed in [23] that residual offset significantly increases at high spin frequencies.

The rest of this paper is organized as follows. In Section 2, we construct an error model for the various offset sources of the Hall plate. Based on this model, we discuss classical spinning, after which we introduce randomized spinning. In Section 3, we introduce offset feedback loops adapted for making them interoperable with randomized spinning, leading to the spread-spectrum offset reduction loop (SS-ORL). The prototype 2 MS/s Hall system enabled by the randomized spinning and the SS-ORLs is showcased in Section 4. Finally, measurements on this prototype are extensively discussed in Section 5.

## 2. Error Model

Integrated Hall plates come in two distinct flavors: horizontal and vertical. Our test chip and drawings mostly hint at horizontal Hall plates. These typically have a $90^{\circ }$ rotational symmetry and four electrical contacts. Vertical Hall plates often have more than four contacts, but usually some contacts are short-circuited or left open, such that in the end only four electrical nodes are used for the readout [24]. Therefore, as far as readout is concerned, vertical Hall plates can also be considered four-terminal devices, see for instance the configurations as detailed in Figure 1c from [24]. So, the error model developed hereafter is valid for both types of Hall plates.

### 2.1. Hall Plate Readout Configuration and Static Offsets

The application of current spinning to a four-terminal device implies that the function of the Hall plate contacts is periodically swapped from biasing to sensing and vice versa. The main readout configurations thus arising are detailed in Figure 1.

These are grouped in two rows, where each row is a representative scheme for four-phase spinning. In the upper scheme, the bias contacts are rotated clockwise (for instance, $V_{\mathrm{DD}}$ is sequentially connected to node 1, 2, 3 and 4), while the sense contacts are rotated counter-clockwise (for instance $V_{\mathrm{HP}-}$ is sequentially connected to node 4, 3, 2 and 1). In the lower scheme, both the bias and the sense contacts are rotated clockwise. Historically, the lower scheme has been applied first [1,10,25], but the upper scheme also has a long track record [2,10,18,26,27]. The advantage of using the upper scheme of Figure 1 is that the readout offset and 1/f noise combines additively with the offset of the Hall plate, and these readout-related error components are therefore also combated by this type of spinning (see Figure 5.33 from [10]). For this reason, the spinning scheme in Figure 1a is our preferred method and the one we will use as the foundation for the rest of our paper. Nevertheless, this choice is not limiting, and the randomized spinning techniques introduced further on are fully applicable to the spinning scheme of Figure 1b as well.

In each configuration, or “phase”, a specific Hall plate voltage $V_{\mathrm{HP}}=V_{\mathrm{HP},+}-V_{\mathrm{HP},-}$ is generated, containing the wanted magnetic signal $V_{S}=S_{I}\cdot B_{⊥}\cdot I_{B}$ (with $S_{I}$ the Hall plate sensitivity in V/T/A and $B_{⊥}$ the magnetic field perpendicular to the plate) and a phase-related offset voltage $V_{o,N}$, where *N* signifies the configuration number. We number the configurations from 0 to 3 for reasons that will become clear later on. As these offsets remain present for as long as the biasing is applied, we refer to them as static offsets.

Since the different readout phases are equivalent (i.e., none of them is preferred or better than the others), it makes sense to take the average of the different offset values:(1)$$V_{o}=\frac{+V_{o,0}+V_{o,1}+V_{o,2}+V_{o,3}}{4}$$

We will refer to $V_{o}$ as the “raw offset” of the Hall plate. Another important number is the offset that remains after averaging over four distinct readout phases:(2)$$V_{r}=\frac{+V_{o,0}-V_{o,1}+V_{o,2}-V_{o,3}}{4}$$

This is a measure of the “residual offset” that remains after four-phase spinning. For a horizontal Hall plate, according to literature the residual offset is typically two orders of magnitude smaller than the raw offset [28,29,30], which is further confirmed by our measurements, see Section 5. For vertical Hall plates, there is more deviation from the ideal behavior, but still the residual offset with four-phase spinning is quite low compared to the raw offset [24].

We now have two offset parameters, $V_{o}$ and $V_{r}$, that both depend on the four distinct offsets that can occur in each configuration. These two offset parameters are more informative about the expected offset behavior during the Hall readout than the individual four offset values $V_{o,N}$. We can now complete the picture by introducing two additional offset-related values: (3)$$\begin{array}{cc} V_{m1} & =\frac{+V_{o,0}+V_{o,1}-V_{o,2}-V_{o,3}}{4} \end{array}$$(4)$$\begin{array}{cc} V_{m2} & =\frac{+V_{o,0}-V_{o,1}-V_{o,2}+V_{o,3}}{4} \end{array}$$

We will refer to these as the two “offset mismatch” parameters because they encode information on how the offset varies depending on the configuration being used. It is now straightforward to show that the four offset quantities ($V_{o}$, $V_{r}$, $V_{m1}$ and $V_{m2}$) defined by Equations (Equation 1)–(4) can be used to replace the four offset values $V_{o,N}$ (N = 0, 1, 2 and 3). This leads to the following expression for the phase-related offsets:(5)$$\begin{array}{c} V_{o},N=V_{o}+c\cdot V_{r}+d\cdot V_{m1}+c\cdot d\cdot V_{m2} \end{array}$$

In this, the coefficients *c*, *d*, and c·d assume only the values +1 and −1, and depend on the readout configuration (i.e., the value of *N*) in the following way:(6)$$\begin{array}{ccccc} N\left(\mathrm{decimal}\right) & 0 & 1 & 2 & 3\\ N\left(\mathrm{binary}\right) & 00 & 01 & 10 & 11\\ c & + & - & + & -\\ d & + & + & - & -\\ c\cdot d & + & - & - & + \end{array}$$

Note that the coefficient *c* is +1 in an even-numbered configuration and −1 in the odd-numbered phases. Therefore, *c* relates to the main axis of the Hall plate bias current flow, e.g., horizontal versus vertical current flow in Figure 1a. The coefficient *d* relates to the direction along this current axis, with d=+1 for up-to-down or right-to-left current flow, and d=−1 for bottom-to-top and left-to-right current flow.

Referring again to Figure 1a, we see that the magnetic signal $V_{S}$ is modulated with the coefficient *c*. Bringing in (Equation 5) as well, the Hall plate voltage $V_{\mathrm{HP}}$ can be written as:(7)$$\begin{array}{c} V_{\mathrm{HP}}=c\cdot V_{S}+V_{o}+c\cdot V_{r}+d\cdot V_{m1}+c\cdot d\cdot V_{m2} \end{array}$$

To obtain an estimate for the magnetic signal $V_{S}$, the Hall readout signal (Equation 7) needs to be demodulated with the *c* chop signal. Using (Equation 7) and the relation $c^{2}=1$, the demodulated Hall signal ${\hat{V}}_{S}$ is obtained as:(8)$${\hat{V}}_{S}=c\cdot V_{\mathrm{HP}},$$
and has the following expression:(9)$$\begin{array}{c} {\hat{V}}_{S}=\underset{\mathrm{not modulated}}{\underset{︸}{V_{S}+V_{r}}}+\underset{\mathrm{modulated}}{\underset{︸}{c\cdot V_{o}+c\cdot d\cdot V_{m1}+d\cdot V_{m2}}} \end{array}$$

The above derived equation is valid for every (binary) value of *c* and *d*, see also (Equation 6). Therefore, it is valid for each of the configurations of Figure 1. In a spinning scheme we will cycle through these configurations (corresponding to making *c* and *d* time-dependent). The goal of such a spinning scheme is to make the error components vanish or at least make them harmless. By inspection of (Equation 9) it is, however, clear that one error component, $V_{r}$, remains unmodulated by the spinning signals *c* and *d* and is indistinguishable from the signal $V_{S}$. Hence, such a spinning scheme is unable to eliminate this $V_{r}$ component. For this reason, we consider $V_{r}$ as the fundamental offset suppression limit of such a spinning scheme.

The offset components discussed so far are all expected to be proportional to the bias current (or bias voltage) of the Hall plate. In practice, there are additional mechanisms, mostly linked to the readout chain, that introduce similar offset components which are independent of the Hall plate bias current. We already discussed that the offset and 1/f noise of the readout chain can be incorporated into $V_{o}$. Likewise, mismatched clock feedthrough and charge injection can lead to independent components that are similarly modulated as $V_{r}$, $V_{m1}$ and $V_{m2}$.

### 2.2. Classical Four-Phase Hall Plate Spinning

In a classical four-phase spinning scheme, the Hall plate phases change in a regular pattern, going from left-to-right and then back to the beginning. Each phase occupies a fixed time slot. The duration of a spin period is denoted as $T_{\mathrm{spin}}$. The coefficients *c*, *d* and c·d in (Equation 5) then change over time. Because the coefficients in (Equation 5) are ±1, each of the terms $V_{r}$, $V_{m1}$ and $V_{m2}$ are modulated with a ±1 sequence, in particular with c(n), d(n) and c(n)·d(n), respectively, with *n* indicating the time slot being addressed. These three modulation sequences are plotted in Figure 2 for the considered classical four-phase spinning scheme from Figure 1a.

It is clear that c(n) represents a classical chopping square wave with period $2T_{\mathrm{spin}}$. The term $c\cdot V_{o}$ in (Equation 9) represents the raw offset being up-modulated to the chop frequency. Observing *d* and c·d in Figure 2 shows that these are both regular chop signals having a period $4T_{\mathrm{spin}}$. So, the terms $c\cdot d\cdot V_{m1}$ and $d\cdot V_{m2}$ in (Equation 9) explain the appearance of half-rate chop signals. The aforementioned offset tones are also shown in a simulated spectrum in Figure 3, where their magnitudes are based on our measurement results. Both Figure 2 and Figure 3 show additional tones which will be addressed later on.

### 2.3. Randomized Four-Phase Hall Plate Spinning

It is clear that the modulating functions c(n), d(n) and c(n)·d(n) are a direct result of the chosen sequence of spin phases. We now reverse this process. Suppose we have two particular modulation sequences c(n) and d(n) in mind, can we derive from these the needed sequence of readout configurations? The answer to this is affirmative. This can be seen by closer examination of (Equation 6). To make the link explicit, we have added the binary representation of the configuration number. It can be seen that the coefficient *c* only depends on the LSB of the binary phase number, while the coefficient *d* is fully determined by the MSB of the phase number. This means that there is a one-to-one correspondence between a set of arbitrary c=±1 and d=±1 values on the one hand and the configuration number of the readout on the other. This correspondence is also why we have numbered the phases starting from 0. Note that after fixing the values for *c* and *d*, the c·d modulation function cannot be assigned independently but is obtained from the product of *c* and *d*.

We now focus our attention on what would be interesting modulation functions. An excellent consequence of the classical four-phase spinning is that the largest offset value, i.e., the raw offset $V_{o}$, is placed as far away as possible from the signal band (see Figure 3). This is especially advantageous when applying the upper spinning scheme in Figure 1a because then the offset and 1/f noise of the analog front-end are also shifted in the same way as $V_{o}$. From an SNR point of view, it is certainly preferable to keep the 1/f noise away from the signal band. This implies that for c(n) we want to retain the modulation at the fastest possible rate, i.e., as shown in Figure 2. The use of a regular $c\left(n\right)={\left(-1\right)}^{n}$ sequence also implies that the signal is modulated in the classical way.

Turning now to the modulation function d(n), we know that in the classical scheme this modulation function is responsible for the half-rate tone. Therefore, in the present paper we investigate the situation in which we take a randomized signal that approximates white noise as the modulation function *d*. The product modulation function c·d then also translates into white noise. The outcome is that the energy associated with $V_{m1}$ and $V_{m2}$ is spread over the full frequency band and hence no longer appears as a parasitic half-rate tone. Somewhat similar spread-spectrum chopping has been reported in some forms in prior art before. In [31,32], band-limited spread-spectrum chopping is applied to somewhat spread out the energy of spurious signals from chopping and auto-zeroing, while in [33,34,35,36] both bandpass and full spread-spectrum chop clocking has been applied to improve the EMI robustness of Sigma–Delta modulators. However, with four-phase current spinning we have a unique case where a pseudo-random sequence can be used for modulating various offset components, while retaining classical chopping as far as the signal component is concerned.

We now look at what the choice for a regular c(n) and a random d(n) implies for the readout configuration being used. An alternating c(n) means that an even phase is always followed by an odd phase and vice versa. A random choice for *d* then means that, when in an even phase, we randomly choose between phase 0 (d=+1) or 2 (d=−1), and when in an odd phase, we randomly select phase 1 (d=+1) or 3 (d=−1). For the rest of the paper, the combination of a regular *c* with a random *d* will be referred to as randomized four-phase spinning.

### 2.4. Dynamic Spinning Effects

Until now, we considered the quasi-static offset components associated with the different Hall plate readout configurations and offset (and 1/f noise) associated with the readout chain. There are also, however, dynamic mechanisms that translate into offset.

It is known that, especially at higher spinning frequencies, dynamic errors can also act as a source of offset. The origin of these errors lies in the parasitic capacitances that exist at each of the Hall plate nodes. At any time, two of these capacitances are charged in accordance with the bias voltage. Before these nodes can be used as readout nodes (which is needed both in the classical as well as in the randomized spinning schemes), the charge associated with the bias-voltage needs to be evacuated. Due to the finite resistivity of the plate, this happens with a finite time constant. In principle, the dynamic error can be reduced to a large extent by zero-banding (ZB) the Hall signal during these transients. However, zero-banding also deteriorates the noise performance of the readout [13]. Moreover, for higher bandwidths, $T_{\mathrm{spin}}$ becomes smaller, and the zero-banding time can no longer be taken to reach full settling. Moreover, mismatch in the spinning and chopper switches results in dynamic errors. It is well known that with traditional four-phase spinning, and in fact in any traditional chopping amplifier circuit, the error spikes translate into offset through a mechanism referred to as spike demodulation [37]. Here we model the phenomenon in a broader context that also covers randomized spinning.

The mechanism is illustrated in Figure 4 in the situation where randomized spinning is applied. The figure shows some illustrative waveforms of the voltages at pin 1 and 2 of the Hall plate (see also Figure 1). The voltages on the other pins are not shown, but these are very similar, except that they are flipped over vertically. The spikes that appear on $V_{\mathrm{HP}}$ are displayed at the bottom. There are two distinct situations that are color-coded: when *n* is odd, the spikes’ origin is indicated in red, while for *n* even, it is in blue. Based on the figure, it can be understood that at any instant in time *n*, the sign of the dynamic error depends on two factors: (i) the sign of the bias voltage as used in the previous readout configuration, i.e., in time slot n−1, and (ii) the sign associated with the connection between the Hall readout nodes and the front-end in the current time slot *n*. The bias sign is fully determined by the value d(n−1), while the readout sign is given by d(n). The effect is multiplicative, and so the sign of the error is d(n)d(n−1). The magnitude of this dynamic error is determined by a certain value, which we denote as $V_{D,1}$. The additional error is therefore given by $d\left(n\right)d\left(n-1\right)V_{D,1}$.

The electrical RC effect as discussed above is expected to fully die out within one spin interval. Nevertheless, we have detected in our measurements relatively small offset mechanisms that extend over a few spin intervals. It is actually because we use randomized spinning that we can differentiate these offset sources from those which were already expected a priori. These offsets lead to terms $d\left(n\right)d\left(n-k\right)V_{D,k}$ with *k* an odd integer larger than 1, and $V_{D,k}$ representing the magnitudes of these extra offset-related components. We will refer to all terms k≤1 as “dynamic offset errors”, albeit that at the time of writing the present paper, we do not have a full understanding of the origin of the extra terms beyond k=1. This does not pose a major issue, however, because the concept of randomized spinning relies only on having a *model* of the relevant offsets that appear, after which the techniques described further on can be applied to remove them.

At this point, we have discussed the main offset sources and have built a model that shows how these different offsets are modulated by a classical or a randomized spinning scheme. We can now add the dynamic error sources to our previous expression (Equation 9) for the demodulated Hall signal, leading to:(10)V^S(n)=VS(n)+Vr+c(n)Vo+c(n)d(n)Vm1+d(n)Vm2=+∑k=1,koddKc(n)d(n)d(n−k)VD,k︸dynamicerrors

For the situation with traditional four-phase spinning, we already showed the modulation functions that up-modulate $V_{o}$, $V_{m1}$ and $V_{m2}$ in Figure 2. We added to this c(n)d(n)d(n−1) and c(n)d(n)d(n−3), which represent the functions modulating the two largest dynamic errors. It is clear that with the traditional spinning scheme from Figure 1a both these dynamic errors appear at DC and hence form additional sources of residual offset beyond the fundamental $V_{r}$ component (see also Figure 3, where all considered dynamic errors map to DC). In the next section, we discuss that with randomized spinning the dynamic errors can be kept separated from the residual offset.

### 2.5. Modulations Obtained with Randomized Spinning

The main difference between traditional and randomized spinning is that in the former case the modulation function *d* is a half-rate square wave, while in the latter case it is a random sequence. We limit ourselves here to random sequences with a flat spectral density. An example of the modulation functions appearing in the case of randomized Hall plate spinning is shown in Figure 5. The example output spectrum in Figure 3 that applies for classical four-phase spinning has now been replotted with *d* generated by a maximum-length pseudo-random sequence. The result is shown in Figure 6. It is clear that the offset terms $V_{m1}$, $V_{m2}$ and all $V_{D,k}$ from (Equation 10) are now modulated into *pseudo-random* signals that appear white. The only offset remaining at DC is the residual offset $V_{r}$. Because the randomized spinning scheme has transformed all offset terms $V_{D,k}$ into pseudo-noise, the offset behavior is vastly improved. At this point, however, it is clear that this comes at the cost of a significantly higher noise floor. This is because the offset mismatch terms $V_{m1}$ and $V_{m2}$ and the dynamic offset errors $V_{D,k}$ are still present, except they have now been distributed over the full Nyquist band. We will refer to these terms as the randomized offset components. The remaining task is therefore to show how these randomized offset components can be reduced and even fully removed by means of spread-spectrum offset reduction loops (SS-ORLs).

### 2.6. System-Level Overview of the Proposed Offset-Suppression Solution

Figure 7 presents a system-level overview of the proposed solution. On the left side shown in green, we have visualized the error model that we constructed in Section 2.1 (see also (Equation 7)), with the dynamic offset errors added from Section 2.4. As explained before, for the modulation function c(n) a regular modulation signal is chosen, i.e., $c\left(n\right)={\left(-1\right)}^{n}$. All paths with a factor d(n) represent the randomized offset components.

The middle part of the figure (with red background color) represents the Hall readout chain. Note that in our initial analysis we demodulated $V_{\mathrm{HP}}$ to obtain an estimate of the signal, i.e., ${\hat{V}}_{S}=c\cdot V_{\mathrm{HP}}$ as given by (Equation 8). In reality, $V_{\mathrm{HP}}$ is first fed into the readout chain and is only demodulated afterwards. To complete the analysis, the dynamic behavior of the readout chain needs to be taken into account. The output of the readout chain, which we denote as $V_{\mathrm{RO}}$, can now be written as:(11)$$V_{\mathrm{RO}}=T\left\{V_{\mathrm{HP}}\right\}$$

In this, *T* denotes the response of the readout chain. The $T\{V_{\mathrm{HP}}\}$ notation needs to be interpreted as the readout chain acting upon the signal $V_{\mathrm{HP}}$. As a first approximation, the response *T* can be modeled as a simple static gain *A*. For a more accurate model, however, the dynamic filter effects that occur in the readout chain should be included as well. In the prototype Hall sensor described in Section 4 of this paper, we employ a readout chain with a digital output. Therefore, all ORLs shown in blue in Figure 7 are implemented fully digitally, which reduces the cost in power and chip area for the SS-ORLs. Furthermore, as will be discussed later on in Section 3.2, the integrator α/(z−1) is a vital part of the SS-ORL system. In particular, the value of the corresponding α coefficient should be carefully designed based on specifications such as noise and start-up time to ensure proper performance of the SS-ORLs. However, the implementation can also be performed in the analog or mixed-mode domain, where the offset reduction loops are then closed at the input of the readout chain instead. The description of the spread-spectrum ORLs in Section 3 remains valid independent of the applied implementation method.

On the top-right side, an optional post-readout ORL that operates at DC has been added. As mentioned before, the raw offset of the Hall plates $V_{o}$ can be substantial; therefore, the readout chain typically already comprises a means of suppressing this dominant offset component to prevent loss of dynamic range or saturation. However, an analog implementation will typically not succeed in fully removing this term and/or may introduce other offsets that replace the raw offset (e.g., the offset of an analog integrator used in the ORL). In that case, removal of some remaining raw offset by means of post-processing may still be needed, hence the optional ORL. In the mostly-digital ORL of [13], as also used in this work, the integrator is in the digital domain and therefore has ideal offset and infinite DC gain (see also Section 4). In that case, this optional post-readout ORL is not needed.

At the far-right side, the demodulation with c(n) is present to obtain the final, offset-compensated, output signal $V_{\mathrm{out}}$. The remaining parts at the right side of Figure 7 represent the post-readout spread-spectrum offset reduction loop that is proposed in the next section to deal with the randomized offset components.

## 3. Spread-Spectrum Offset Reduction Loops

### 3.1. System Derivation

To derive our system equations, we will now elaborate Figure 7. By inspection of the the figure, we can write down the raw readout output signal $V_{RO}$, defined in (Equation 11), as:$$\begin{array}{cc} V_{\mathrm{RO}}\; & =T\underset{=V_{\mathrm{HP}}}{\underset{︸}{\left\{c\left(n\right)\left(V_{S}+V_{r}\right)+V_{o}+d\left(n\right)V_{m1}+c\left(n\right)d\left(n\right)V_{m2}+\sum_{\begin{array}{c} k=1\\ k\;\mathrm{odd} \end{array}}^{K}d\left(n\right)d\left(n-k\right)V_{D,k}\right\}}} \end{array}$$

Furthermore, assuming the readout chain and therefore also the operator *T* to behave linearly:(12)$$\begin{array}{cc} V_{\mathrm{RO}}\; & =T\left\{c\left(n\right)\left(V_{S}+V_{r}\right)\right\}+\underset{\approx 0}{\underset{︸}{T\left\{V_{o}\right\}}}\\ \; & \;+T\left\{d\left(n\right)V_{m1}+c\left(n\right)d\left(n\right)V_{m2}+\sum_{\begin{array}{c} k=1\\ k\;\mathrm{odd} \end{array}}^{K}d\left(n\right)d\left(n-k\right)V_{D,k}\right\} \end{array}$$

The first term corresponds to the up-modulated input signal and the unavoidable residual offset. This term will be amplified and then leads, after demodulation, to the signal-related component in the output signal. The second term in the equation represents the reaction of the readout chain to the raw offset $V_{o}$. In our prototype (discussed in Section 4 and Section 5), we use the state-of-the-art readout chain described in [13] which has a mixed-mode ORL to force this term to become strictly equal to zero. As discussed in the previous Section 2.6, any good readout chain is likely to have some mechanism to suppress this error component to avoid issues with dynamic range and saturation, which can be further enhanced by the optional ORL operating at DC that is also visible in Figure 7. Hence, in practice, it is normally justified to assume that this term is negligible at the output of the readout chain, which will be assumed in what follows (if this error contribution is not negligible, it will result in a spurious output tone at the chop frequency, as in Figure 6. The analysis of the other offset-related terms in (Equation 12) is unaffected by this). We will now focus on the remaining offset-related terms in (Equation 12).

To start our explanation for the actual offset reduction loops (the blue part in Figure 7), we will for the moment assume that each loop operates in such a way that good estimates of the offset parameters $V_{m1}$, $V_{m2}$ and $V_{D,k}$ are produced at the respective nodes ${\hat{V}}_{m1}$, ${\hat{V}}_{m2}$ and ${\hat{V}}_{D,k}$. We will explain below (at the end of this section) that the loops do exactly this, and we will further discuss the accuracy of the estimates. With this assumption, it makes sense to use the estimates to remove the unwanted offset terms from $V_{\mathrm{RO}}$. This leads to the corrected readout denoted as $V_{\mathrm{ROC}}$, defined as follows:(13)$$\begin{array}{c} V_{\mathrm{ROC}}=V_{\mathrm{RO}}-\hat{T}\{d\left(n\right){\hat{V}}_{m1}+c\left(n\right)d\left(n\right){\hat{V}}_{m2}+\sum_{\begin{array}{c} k=1\\ k\;\mathrm{odd} \end{array}}^{K}d\left(n\right)d\left(n-k\right){\hat{V}}_{D,k\}} \end{array}$$

Here $\hat{T}$ represents an estimate (or replica) of the readout chain transfer function *T*. When put in a block diagram, this leads to the error correction block displayed in Figure 7 with light-blue background color. Notice that the estimated offset parameters are simply modulated with the already discussed randomized modulation functions. Combining now (Equation 12) with (Equation 13), assuming the offset parameters to be quasi-static (relative to the dynamics of the readout chain) and assuming $\hat{T}=T$, we can derive:(14)$$\begin{array}{cc} V_{\mathrm{ROC}}\; & =T\left\{c\left(n\right)\left(V_{S}+V_{r}\right)\right\}\\ \; & \;+\hat{T}\left\{d(n)\right\}\left(V_{m1}-{\hat{V}}_{m1}\right)+\hat{T}\left\{c(n)d(n)\right\}\left(V_{m2}-{\hat{V}}_{m2}\right)\\ \; & \;+\sum_{\begin{array}{c} k=1\\ k\;\mathrm{odd} \end{array}}^{K}\hat{T}\left\{d(n)d(n-k)\right\}\left(V_{D,k}-{\hat{V}}_{D,k}\right) \end{array}$$

If the estimates ${\hat{V}}_{m1}$ and ${\hat{V}}_{m2}$ perfectly match the corresponding offset parameters $V_{m1}$, $V_{m2}$ and $V_{D,k}$, then the second and third lines in the above equation disappear, and the effect of these error terms vanishes resulting in an ideal error correction. However, if they do not match perfectly, it becomes clear from (Equation 14) that $V_{ROC}$ comprises information on how far the estimated offset values deviate from the real values, e.g., $V_{m1}-{\hat{V}}_{m1}$. These estimation errors appear as a scale factor of various signals, e.g., $\hat{T}\left\{d\left(n\right)\right\}$). These signals correspond in their turn to the reconstructed response of the readout chain to one of the randomized modulation functions that we already encountered in the previous sections, e.g., d(n). These responses are therefore by themselves also a random signal. While all the random signals appearing in (Equation 14) are linked to a single random sequence d(n) (and a non-random c(n)), the different random signals in (Equation 14) nevertheless appear to be uncorrelated (while this is not true for any choice of random sequence d(n), in practice, we only need a pseudo-noise sequence d(n) which does provide this result. We propose the use of a maximum-length sequence, which is very easy to implement. The uncorrelatedness property can then be confirmed by means of simulation). By correlating $V_{\mathrm{ROC}}$ with the reconstructed random signals $\hat{T}\left\{d\left(n\right)\right\}$, $\hat{T}\left\{c\left(n\right)d\left(n\right)\right\}$ and $\hat{T}\left\{d\left(n\right)d\left(n-k\right)\right\}$, a measure of the estimation error can be obtained, resulting in:(15)$$V_{m1}-{\hat{V}}_{m1}=\frac{E\left[V_{\mathrm{ROC}}\cdot \hat{T}\left\{d\right\}\right]}{E\left[{\left(\hat{T}\left\{d\right\}\right)}^{2}\right]}$$
and similar formulae for $V_{m2}-{\hat{V}}_{m2}$ and $V_{D,k}-{\hat{V}}_{D,k}$. In this, $E\left[\cdot \right]$ denotes taking the statistical expected value. The denominator represents the power of the random signal. In reality, the correlation in (Equation 15) is implemented as a multiplication with the random signal, i.e., the expectation operator E is dropped. Note that such an implemention for correlating a signal with a known random sequence is very common in analog and mixed-mode circuits with adaptively tuned coefficients [38,39,40,41,42]. Moreover, instead of normalizing with the real power, it is more convenient to normalize with the power that arises when $\hat{T}=A$, with *A* the nominal gain of the readout chain. This then leads to a simple scale factor $1/A^{2}$. With these elements, the correlation block as indicated in Figure 7 arises, and thus we obtain the error signal $e_{V_{m1}}$ as:(16)$$e_{V_{m1}}=\frac{V_{\mathrm{ROC}}\cdot \hat{T}\left\{d\right\}}{A^{2}}$$

From Equation (Equation 15), it is clear that:(17)$$E\left[e_{V_{m1}}\right]=\frac{E\left[{\left(\hat{T}\left\{d\right\}\right)}^{2}\right]}{A^{2}}\left(V_{m1}-{\hat{V}}_{m1}\right)$$

This last equation shows that the correct ${\hat{V}}_{m1}$ is obtained when $E\left[e_{V_{m1}}\right]=0$. The equation $E\left[e_{V_{m1}}\right]=0$ can now be solved using well known stochastic approximation methods that trace back to [43]. When applied to our problem at hand, the method consists of making subsequent estimates of ${\hat{V}}_{m1}\left(n\right)$:(18)$${\hat{V}}_{m1}\left(n+1\right)={\hat{V}}_{m1}\left(n\right)+\alpha_{n}\cdot e_{V_{m1}}$$

In the original paper [43], $\alpha_{n}$ is a sequence that in the limit goes to zero, and it is proven that the estimates converge to the exact value (in our case $V_{m1}$). However, letting $\alpha_{n}$ converge to zero does not make sense in our present context because we want to allow that the true offset parameter $V_{m1}$ can change slowly over time, and so we want to maintain some minimal adaptability of the system. Therefore, we use the above adaptation rule with a fixed adaptation parameter, i.e., $\alpha_{n}=\alpha$. In this case, (Equation 18) boils down to applying the error signal $e_{V_{m1}}$ to the integrator α/(z−1) to obtain the estimated offset parameter ${\hat{V}}_{m1}$. We thus obtain the closed loop system shown in Figure 8.

A more intuitive way to understand this system is that the integrator α/(z−1) adapts the estimated value ${\hat{V}}_{m1}$ until the DC component from its input (the error signal $e_{V_{m1}}$) becomes equal to zero. Therefore, because of (Equation 17), the DC component of the estimated value ${\hat{V}}_{m1}$ becomes equal to that of the actual value of $V_{m1}$ as we assumed at the start of our explanation above. The system thus obtained, which we refer to as the spread-spectrum ORL, provides estimates ${\hat{V}}_{m1}$ of the true value $V_{m1}$. Moreover, it can track changes of this true value over time. It has one key design parameter, i.e., the integration constant α, which can also be viewed as the “adaptation parameter” of the ORL. Its sizing will be discussed later on. The ORLs for the other offset parameters $V_{m2}$ and $V_{D,k}$ are set up in the same way and operate simultaneously, leading to similar loops as the one in Figure 8 that details the loop for estimating $V_{m1}$.

Until now, we disregarded the impact of the input signal, i.e., the first term in (Equation 14). It is, however, clear that the random sequence d(n) can be taken independent from the magnetic input signal; therefore, the correlation of the input-related term and the random signals is expected to be zero. This is the fundamental reason why the SS-ORL can work “under the radar” without notching out signal components passing through the system. However, as explained in the next section, the input signal can have a significant impact on the variance of the correlation terms, and this needs to be taken into account.

### 3.2. Sizing the SS-ORL’s Integration Factor α

In the above, we explained the operation of the SS-ORL in terms of correlations between random signals. However, because in practice one cannot average over an ensemble (there is, after all, only one system), statistical averaging needs to be replaced by averaging over time. This averaging is automatically realized by the integrators that adapt the estimates to the right value. However, even though the estimates ${\hat{V}}_{o}$, ${\hat{V}}_{m1}$, ${\hat{V}}_{m2}$ and ${\hat{V}}_{D,k}$ have exactly the same DC component as their offset term counterparts from (Equation 10), they do have a non-zero variance. This is because, due to the presence of the input signal, the signals coming from the correlation block (i.e., the error signals $e_{V_{m1}}$, $e_{V_{m2}}$, …) can exhibit a large variance. As a result, the integrator outputs ${\hat{V}}_{o}$, ${\hat{V}}_{m1}$, ${\hat{V}}_{m2}$ and ${\hat{V}}_{D,k}$ drift around the correct values with some finite variance as well. This in its turn leads to incomplete cancellation of the randomized error components and a degradation of the overall system’s noise performance.

While due to the presence of the possibly large input signal we have little control over the variance of the correlation signals, the variance of the offset parameter estimates does depend on the adaptation parameter α and can be scaled down using low α values. On the other hand, a low value of α also reduces the adaptation speed of the SS-ORLs, which is especially problematic at start-up conditions when the initial offset parameter estimates deviate most from the actual values. Therefore, the sizing of the adaptation parameter α involves a trade-off between additional noise and start-up time. It is this trade-off which we try to quantify next.

#### 3.2.1. Noise of the SS-ORLs

Let us first study how the variance of the offset parameter estimates depend on α. To this end, assume that the estimates are set to their correct values at time n=0. We only enable the ${\hat{V}}_{m1}$ loop with a nonzero α, while the other loops are kept frozen (by setting their adaptation parameters to zero). Moreover, since we are only interested in the variance of the estimate, we can assume without loss of generality that the true offset parameter $V_{m1}$ is zero. We are now interested in the variance of ${\hat{V}}_{m1}\left(n\right)$ after the SS-ORL integrator output has settled, i.e., for n→∞:(19)$$\begin{array}{c} \sigma_{m1}^{2}=\lim_{n\to \infty }E\left[{\hat{V}}_{m1}^{2}\left(n\right)\right] \end{array}$$

As a first step, we try to derive an expression for ${\hat{V}}_{m1}$ as a function of $V_{\mathrm{RO}}$. This requires solving the SS-ORL loop, which is complicated by the presence of $\hat{T}$. To simplify matters, we again replace $\hat{T}$ by the overall gain *A* of the readout chain. This approximation is certainly reasonable for the readout chain we implemented in the prototype (see Section 4) because of the nearly full bandwidth that is realized. For other readout chains that exhibit more low pass filtering, the derivation that follows will rather provide an upper limit to the noise in the SS-ORL. By replacing $\hat{T}$ by a gain *A*, the following can be derived:(20)$$\begin{array}{c} {\hat{V}}_{m1}\left(n\right)=\frac{\alpha }{A}\sum_{k=0}^{n-1}{\left(1-\alpha \right)}^{n-k-1}V_{\mathrm{RO}}\left(k\right)d\left(k\right) \end{array}$$

The pseudo-random sequence d(k) is taken to be white, thus having the correlation property $E\left[d(k)d(l)\right]=\delta \left(k-l\right)$. Also using the statistical independence of *d* and $V_{\mathrm{RO}}$, it can be derived that:(21)$$\begin{array}{c} E\left[{\hat{V}}_{m1}^{2}\left(n\right)\right]=\frac{\alpha^{2}}{A^{2}}\sum_{k=0}^{n-1}{\left(1-\alpha \right)}^{2(n-k-1)}E\left[V_{\mathrm{RO}}^{2}\left(k\right)\right] \end{array}$$

Assuming the readout signal $V_{\mathrm{RO}}$ can be modeled as a stationary random signal, then $E\left[V_{\mathrm{RO}}^{2}\left(k\right)\right]$ is independent of *k*, hence a constant factor. Therefore, (Equation 21) reduces to the summation of a geometric series. Taking the limit for n →∞, we obtain a closed expression for the variance (Equation 19):(22)$$\begin{array}{c} \sigma_{m1}^{2}=\frac{1}{A^{2}}\frac{\alpha }{2-\alpha }E\left[V_{\mathrm{RO}}^{2}\right]≃\frac{\alpha }{2A^{2}}E\left[V_{\mathrm{RO}}^{2}\right] \end{array}$$

The approximation on the right side is valid because in practice we always have α≪1. The random variation of the estimate ${\hat{V}}_{m1}$ around its true value $V_{m1}$ translates into extra noise contributions in the compensated readout, as can be seen from (Equation 14). Because the estimate ${\hat{V}}_{m1}$ is modulated by the *d* sequence having unit power, the random error on the corrected readout value $V_{ROC}$ has approximately the variance $A^{2}\sigma_{m1}^{2}$. This extra noise power in the compensated readout $V_{\mathrm{ROC}}$ is not affected by the demodulation with c(n), so this is also the extra noise power in the final (offset-compensated) output signal.

Until now, we considered only the SS-ORL that estimates $V_{m1}$ and evaluated the extra noise power that appears in the offset-compensated output signal. It can be shown now that each of the other SS-ORL loops contribute exactly the same amount. While these other SS-ORLs have a random sequence different from d(n), they also behave as white noise, have unit power and are independent of the signal. Moreover, all modulation sequences are orthogonal to each other. Hence, the total noise power introduced by all SS-ORL loops in the output signal is:(23)$$\begin{array}{c} \sigma_{\mathrm{ORLs}}^{2}\approx \frac{\alpha }{2}\cdot \#\mathrm{ORLs}\cdot E\left[V_{\mathrm{RO}}^{2}\right] \end{array}$$
in which #ORLs denotes the number of active SS-ORLs. From this, it is clear that the worst condition occurs when $|V_{\mathrm{RO}}|$ is large, that is, when the readout value is close to its extremes, i.e., when the magnetic input signal is large.

#### 3.2.2. Start-Up Time of the SS-ORLs

We now calculate the start-up time to reveal its dependency on α. For this, we consider the system of Figure 7 with signal and all offset sources zero except $V_{m1}$. We again assume that the readout chain *T* can be approximated as a uniform gain *A*. Then we have that $V_{\mathrm{RO}}\left(n\right)\approx AV_{m1}d\left(n\right)$. If we now assume that only the ${\hat{V}}_{m1}$ adaptation loop is enabled, the response is fully determined by (Equation 20), which for the particular $V_{\mathrm{RO}}\left(n\right)$ reduces to:(24)$$\begin{array}{cc} {\hat{V}}_{m1}\left(n\right)\; & =\frac{\alpha }{A}\sum_{m=0}^{n-1}{\left(1-\alpha \right)}^{n-m-1}AV_{m1}d^{2}\left(m\right)\\ \; & =V_{m1}\left[1-{\left(1-\alpha \right)}^{n}\right] \end{array}$$

If we now define the start-up time as the time needed to reach 90% of the final value, the above expression leads to:(25)$$\mathrm{start}-\mathrm{up}\;\mathrm{time}=\frac{\ln(1-0.9)}{\ln(1-\alpha )}T_{s}\approx \frac{2.3}{\alpha }T_{s}$$

#### 3.2.3. Discussion

It is clear from (Equation 23) that the additional noise from the SS-ORLs can be made arbitrarily low by choosing a sufficiently small value of α. A possible criterion for selecting an appropriate α value is to make sure that the worst-case extra noise predicted by (Equation 23) is well below the intrinsic noise that is already present at the output of the readout chain, for instance, noise from the Hall plate and the readout front-end. Another option is to use the same criterion but monitor the RMS value of $V_{\mathrm{RO}}$ for determining α. The latter allows for the same noise budget to operate the SS-ORLs with a higher α when the RMS signal level turns out to be low. Yet another option is to add extra low pass filters $H_{\mathrm{LP}}$ in the correlator block. The rationale here is that the input signal is present in $V_{\mathrm{ROC}}$ as an up-converted signal around $f_{\mathrm{chop}}$. Lowpass filtering $V_{\mathrm{ROC}}$ prior to feeding this into the correlator therefore suppresses much of the input signal band. The reconstructed random signals can then also be filtered by the same $H_{\mathrm{LP}}$. It can be shown that these actions do not change the sign of the expected value of the correlation outputs, so adaptation still occurs in the right direction, but they can help in reducing the variance of the correlation signals. This again means that the α value can be larger for the same excess noise level.

In any case, the α value determined from the available noise budget can turn out to be very small, leading to long start-up times as evidenced by (Equation 25). However, it is clear that during start-up, the estimates are so far off from their final value that the noise is much larger than the target noise level anyway. Therefore, it makes more sense to initially use a relatively large value of α for a fast ORL start-up, after which α can be dropped to the lower value needed for optimal noise performance. An example of such a multiple-α algorithm will be demonstrated during the measurements in Section 5.

## 4. The 2 MS/s Hall System Prototype

In the previous sections, we provided a basis for the randomized spinning concept and the SS-ORLs that allow us to minimize the system offset. These offset compensation techniques are to a large extent orthogonal to the actual readout chain, which is why in Figure 7 only the transfer function *T* is needed as a characteristic feature of the readout dynamics. The goal of the rest of the paper is to fully demonstrate the enabling capabilities of these concepts to obtain a high-bandwidth low-offset Hall readout system that provides a 2 MS/s digital output. For this, a prototype has been built consisting of a custom test chip and digital logic to control the readout process. Most of the digital functions are implemented in an FPGA to provide maximum development flexibility. The custom test chip is identical to what has been described in [13], with an important difference being that the clock frequency has been increased by a factor of two and the spin phase is controlled externally. We therefore expect the already high bandwidths reported in [13] to be effectively doubled, with the randomized spinning being instrumental in removing the half-rate spurs as well. Since at these higher clock rates dynamic errors increase in relevance significantly, randomized spinning and the SS-ORLs become important tools to also suppress these dynamic errors and maintain a low-offset behavior in spite of the higher spinning/chopping speeds.

A simplified block diagram of the implemented readout chain is shown in Figure 9, while chip level implementation details for the contents of the prototype chip are displayed in Figure 10. Four Hall plates form the magnetic sensor. These four plates are hardwired in parallel to lower the fundamental noise limit posed by the sensing structure. The weak Hall voltage is amplified by an “In-Loop Sampling Amplifier” (ILSA). As the name implies, the distinctive feature of this readout amplifier is the fact that it comprises a sample-and-hold operation inside the amplifier feedback. Advantages of the ILSA are inherent anti-alias behavior, with a filtering characteristic locked into a shape that maximally prevents aliasing towards DC, low noise and an effective bandwidth that is close to the Nyquist frequency [13]. Also notable is the one-step high gain that is obtained, which for the present prototype is 500×. The ILSA itself contains four major parts. The low-noise transconductance (LNT) with value $G_{\mathrm{LNT}}$ is implemented as a super source follower [44], exhibiting improved linearity and lower input capacitance compared to a simple transconductance. A current-to-voltage integrator then feeds the chopped sample and hold block (S&H), in which the integrator output is first sampled onto capacitors $C_{S}$ and then held within the feedback of an opamp. Switch control is such that the output of the S&H is chopped. The last part is the feedback transconductance $G_{\mathrm{FB}}$. The DC gain of the ILSA is defined ratiometrically as $A_{\mathrm{DC}}=G_{\mathrm{LNT}}/G_{\mathrm{FB}}$. To reduce the size of the feedback resistor, a smaller resistor $R_{\mathrm{FB}}/5$ in combination with a down-scaling 1/5 current mirror is used. Because of the inherent anti-aliasing, a simple Nyquist-rate ADC can be put after the ILSA without additional filtering. The digital output *D* then directly forms the input to our system of SS-ORLs, i.e., *D* = $V_{\mathrm{RO}}$ in Figure 7. Also added to the system is a mostly digital ORL, employing a bilinear integrator and current DAC, to remove the $V_{o}$ offset. Advantages of this ORL are: (i) no effect of the ORL on the overall system’s DC gain, (ii) fast start-up, and (iii) no residue of $V_{o}$ in the digital output *D* [13]. The latter advantage makes our assumption $V_{o}\approx 0$ from (Equation 12) absolute.

Figure 11 shows a photo of the 2 MS/s readout prototype. As explained above, the test chip is connected to an FPGA board, communicating with each other through serial LVDS communication lines. A maximum-length linear-feedback shift register ML-LFSR using an N=20 bit shift register is programmed into the FPGA to produce the pseudo-random d(n) sequence. An N-bit LFSR register can produce maximum length sequences that only repeat after $2^{N}-$1 values, and the autocorrelation property of these sequences is known to be $E\left[d(n)d(n-k)\right]=1/\left(2^{N}-1\right)$ for k≠0. The alternating sequence $c\left(n\right)={\left(-1\right)}^{n}$ is also generated in the FPGA. The sequences c(n) and d(n) are sent by the FPGA to the test chip via the LVDS lines, thus controlling the unique spin configuration during the Hall plate readout. This flexibility allows the readout chip to operate with both traditional and randomized spinning.

## 5. Experimental Verification

### 5.1. Measurements with Noise-Constrained α Value

First of all, in the situation where traditional spinning is applied at 2 MHz, the input-referred noise of the complete sensor system has been measured to be 47 μTrms. The measured noise spectrum is flat up to 820 kHz. Above this frequency, the noise is somewhat increased because of upmodulated 1/f noise. The measurements we report here are for the best-noise case, where we use a digital LP filter with bandwidth 820 kHz to remove some excess 1/f noise. The noise value of 47 μTrms is the intrinsic RMS noise level of the readout. Note that to make noise figures easier to compare with analog-only Hall readout systems, we report noise levels as the equivalent input-referred magnetic noise. However, because our noise performance figures are always determined based on the digital output of the sensor system, they represent a more complete picture of the true performance because they include noise-aliasing effects arising from sampling and quantization noise caused by the digitization.

Next, randomized spinning is enabled with the SS-ORLs disabled. A 64 kHz magnetic signal is generated by a PCB coil underneath the chip with amplitude 780 μT. The resulting output spectrum is shown in black in Figure 12. The output noise level is now increased to 250 μTrms. This noise increase is expected because now the energy which was originally present in offset-related spurs is transformed into noise.

We now enable the SS-ORLs to remove the randomized offset terms. We have five loops in total to cover the $V_{m1}$, $V_{m2}$, $V_{D,1}$, $V_{D,3}$ and $V_{D,5}$ components. To calculate the proper value for α, we enforce the condition that the output RMS noise level should never increase more than 20% when enabling the SS-ORLs. As the readout chain has a maximum input range of ±10.6 mT, we have the worst-case value $E\left[V_{\mathrm{RO}}^{2}\right]=0.8^{2}$. Using (Equation 23), the required α value is calculated to be 3.77 $\times 10^{-6}$. To simplify the digital implementation we take the closest power of two. Then this multiplication can be performed by a simple bit shift. This leads to the implemented value α = 1/$2^{18}$. Once the SS-ORLs settle, the output spectrum as shown in red in Figure 12 is achieved. This provides an output RMS noise of 47 μTrms, i.e., exactly the same as the minimum noise level determined earlier. Note that in this experiment, we do not get any increase of the noise level. This is because the input signal level is substantially below the maximum range due to the limitations of our measurement setup (i.e., the PCB coil can only generate magnetic fields of limited magnitude).

Using a magnet, a DC magnetic field can be applied to the chip up to full scale. Figure 13 shows (in red) the measured input-referred RMS noise level divided by the measured intrinsic input-referred RMS noise level plotted as a function of the input DC magnetic field level. As expected, the noise level increases slightly due to the large input signal adding additional noise into the SS-ORLs. By implementing the α as calculated above, the additional noise never adds up to more than a 20% increase of the total input-referred RMS noise level (the 20% threshold is shown in black in Figure 13).

When the input frequency is increased to 810 kHz, i.e., near the end of the bandwidth, the spectrum shown in Figure 14 is obtained. In this, a mirror tone of the input frequency caused by direct inductive coupling between the excitation coil and the Hall plate wiring loops has been removed in the same way as described in [13]. Moreover, the randomized spinning and SS-ORLs loops remove all avoidable offset contributions, leading again to the minimum possible RMS noise level 47 μTrms. This measurement demonstrates both the large-bandwidth capability of the system and the effectiveness of randomized spinning also for signals at the high end of this large frequency range.

### 5.2. Measurements with Stepped α Values

If we plug the noise-constrained α value used above into (Equation 25), a start-up time of 0.3 s is predicted. During this time, the offset-parameter estimates deviate more than 10% from their correct value; hence a substantial noise floor is expected in this time interval. To alleviate this, a multiple-α algorithm is implemented in the FPGA. Using (Equation 25), an α value can be select to achieve a very fast start-up of 1 ms. We obtain $\alpha =1/2^{10}$. The multiple-α algorithm consists of stepping the adaptation parameter as $\alpha =1/2^{n}$, where *n* is stepwise increased from its initial value 10 to 18. Figure 15 shows a measurement result of this technique where a 500 Hz input signal is present.

### 5.3. Offset Histograms

The randomized spinning scheme allows the identification of many more offset components compared to what is possible with a traditional spinning scheme. We already discussed Figure 3 which clearly shows that with traditional spinning all dynamic errors map to DC, where they overlap with the residual offset $V_{r}$. This is in contrast to the SS-ORLs of Figure 7 which can provide an individual estimate of these offset components. However, the most important offset is the residual offset that remains at DC. Therefore, to accurately verify the performance of the randomized spinning techniques, the Hall plate residual offset $V_{r}$ was isolated from other DC offset sources for the measurements below. This can be performed by using a zero-bias Hall plate measurement (as was done in this work) or alternatively by shorting the input to the readout chain.

To perform statistics on the Hall plate offset terms, 10 samples of the Hall readout chip were measured with the ORLs disabled. A normal distribution was fitted over each offset tone’s measured magnitude. The resulting mean μ and standard deviation σ values are given in Table 1. Clearly, $V_{r}$ is much smaller than the other static offset terms. The measurements confirm that $V_{o}$ and $V_{r}$ differ over two orders of magnitude, as stated before. The dynamic offset terms $V_{D,k}$ start in the same order of magnitude as $V_{m1}$ and $V_{m2}$ and drop exponentially for larger *k*. Figure 16 shows how $V_{D,1}$ is the dominant term, as this term embodies the direct RC effect of the spinning transients, which dies out within one spin phase. The terms $V_{D,k}$ for k>1 are much smaller. They show a transient spanning multiple spin phases but only for the odd numbers of *k*. Clearly some slow settling effect is present in the Hall plates of the chip, but as stated before, at the time of writing this paper, we do not have a full understanding of where it stems from.

Figure 17 illustrates the power of randomized spinning as an enabler for low offset. Here, measurements with traditional four-phase spinning at 2 MHz are compared to randomized spinning at the same spin frequency. The residual Hall plate DC offset has dropped from 130 μT to only 23 μT. Figure 18 summarizes a more extensive set of measurements calculated at three different spin frequencies (666 kHz, 1 MHz and 2 MHz). As expected, with traditional spinning the residual offset becomes dominated by the dynamic error terms $V_{D,k}$ as the spin frequency increases. When using randomized spinning, however, the residual Hall plate offset does not change when increasing the spin frequency because only $V_{r}$ remains (which is independent of the spin frequency).

An overview of some important results is showcased in Table 2, compared to other Hall sensors. Using the randomized spinning techniques, we were able to double the spinning frequency and also the signal bandwidth compared to our first verification of the ILSA prototype chip in [13], without increasing the offset or the noise (by a numerical coincidence, the systematic electrical readout offset and the Hall plate dynamic offset error in [13] have the opposite sign. Due to this, in [13] both components partially cancel each other so that the overall offset of [13] is comparable to the results in the present paper, where the dynamic offset error is removed by the SS-ORLs). Furthermore, the signal bandwidth is now free of any in-band offset tones that would appear due to traditional four-phase spinning.

## 6. Conclusions

While traditional four-phase spinning provides great potential to minimize the residual offset of a Hall sensor in practice through the process of spike demodulation, the actual residual offset can increase significantly when employing a high spin frequency. Furthermore, there are offset-related spurious tones at half the chopping frequency which limit the available “tone-free” bandwidth or necessitate calibration techniques for removal thereof. We have shown that randomized spinning circumvents these challenges, as such acting as an enabler for high frequency Hall plate spinning and therefore higher bandwidth Hall sensors without in-band spurious tones while at the same time reducing the residual offset. This conclusion is fully supported by the presented Hall readout system featuring a 2 MS/s digital output rate, a spurious-free bandwidth from 820 kHz (noise-optimal) to 980 kHz (maximum bandwidth) and an input-referred Hall plate offset as low as 23±22 μT.

## Figures and Tables

**Figure 1 sensors-22-06069-f001:**
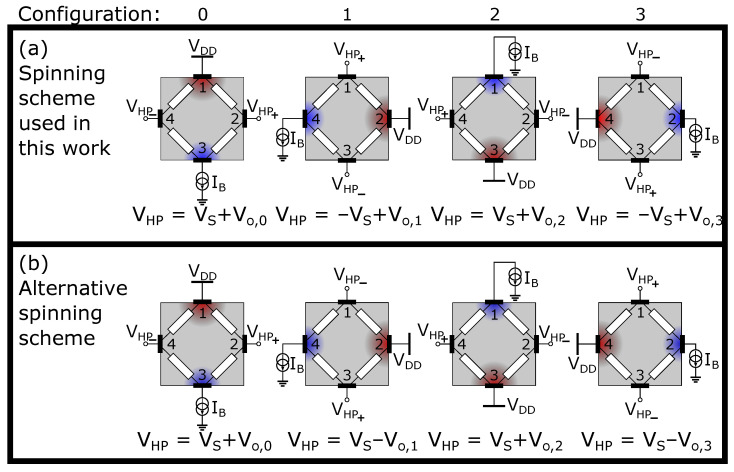
Different readout configurations or “phases” of a Hall plate, assuming current-mode biasing and voltage-mode sensing. Four-phase spinning consists of repeatedly cycling through the configurations on a single row from left to right. The upper scheme results in up-modulation of the magnetic signal $V_{S}$, while the lower scheme up-modulates the offset.

**Figure 2 sensors-22-06069-f002:**
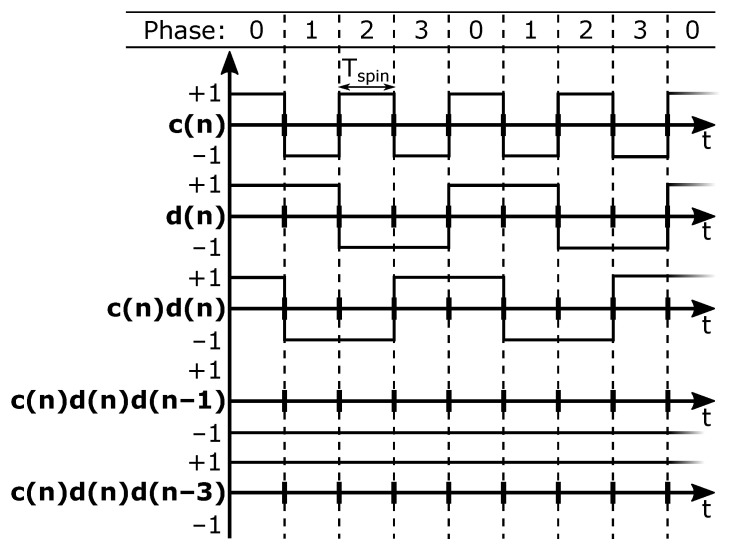
Modulation functions explaining the offset tones originating from classical four-phase spinning.

**Figure 3 sensors-22-06069-f003:**
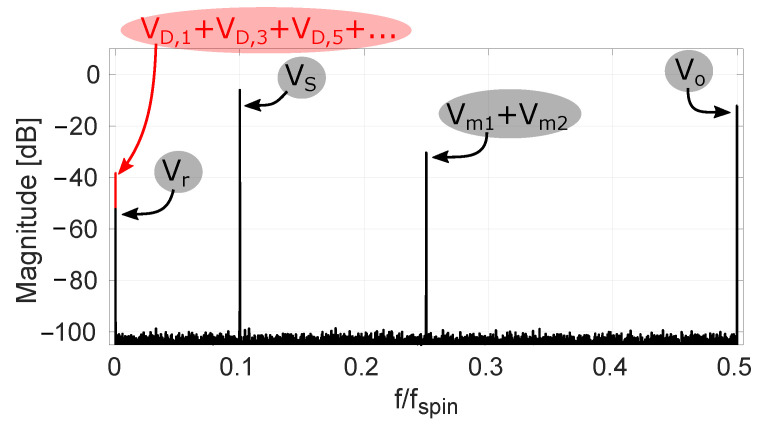
Simulated spectrum with a −6 dBfs input sine wave at frequency 0.1 $f_{\mathrm{spin}}$ for $V_{S}$. Offset tones, corresponding to traditional four-phase spinning, and white noise are present with magnitudes based on our measurements.

**Figure 4 sensors-22-06069-f004:**
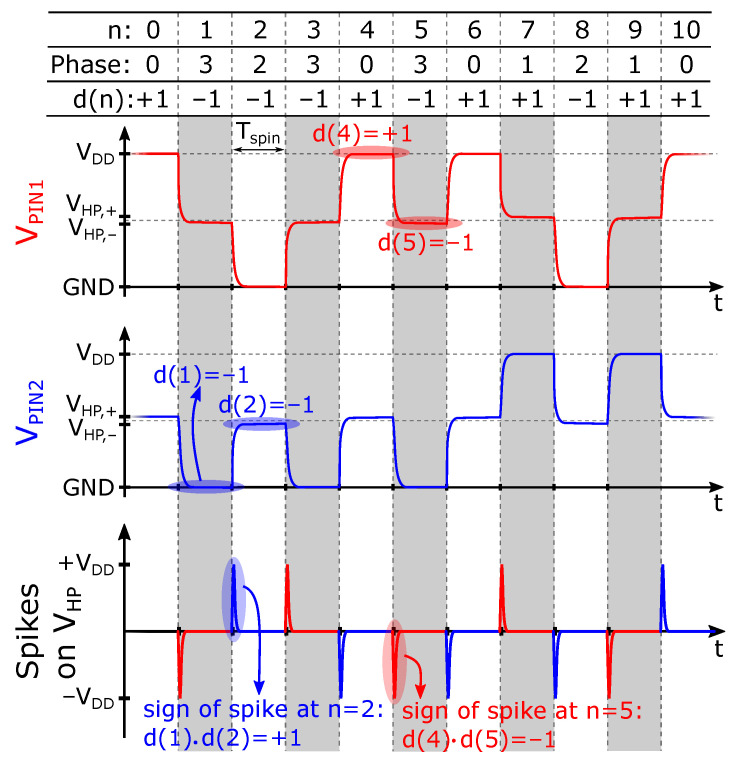
Voltage at Hall plate pin 1 (red) and pin 2 (blue) plotted in time in the case of randomized four-phase spinning. The transient spikes due to Hall plate spinning are also plotted. The phases shaded in grey correspond to odd configuration numbers.

**Figure 5 sensors-22-06069-f005:**
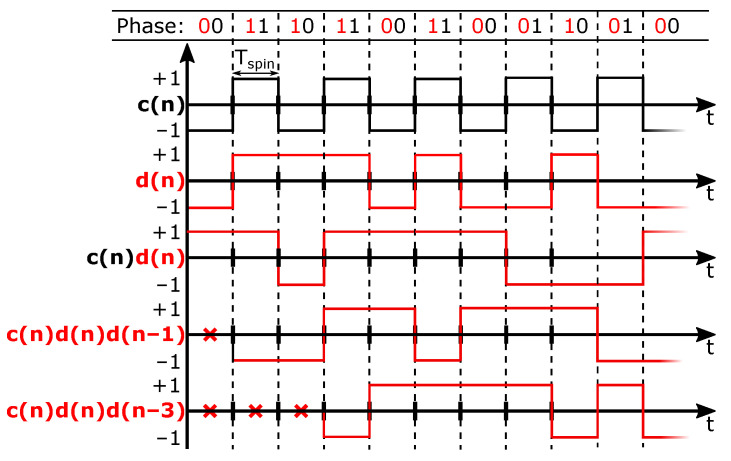
Example of modulation functions resulting from randomized four-phase spinning.

**Figure 6 sensors-22-06069-f006:**
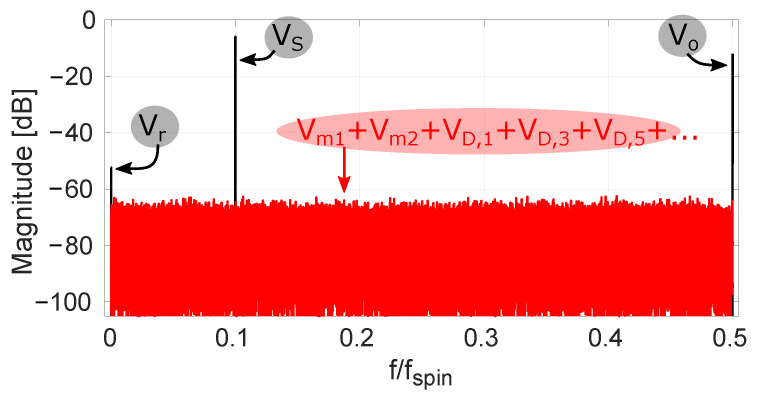
Simulated spectrum with a −6 dBfs input sine wave at frequency 0.1 $f_{\mathrm{spin}}$ for $V_{S}$. Offset tones, corresponding to randomized four-phase spinning, are present with magnitudes based on our measurements.

**Figure 7 sensors-22-06069-f007:**
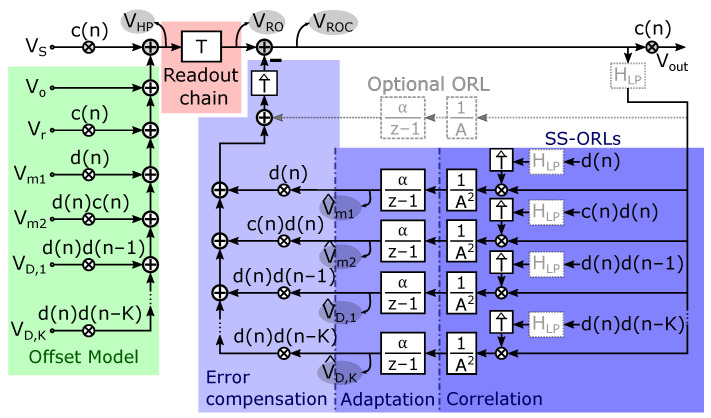
System-level overview of the proposed offset-suppression solution, with various spread-spectrum ORLs to remove the randomized Hall plate offsets.

**Figure 8 sensors-22-06069-f008:**
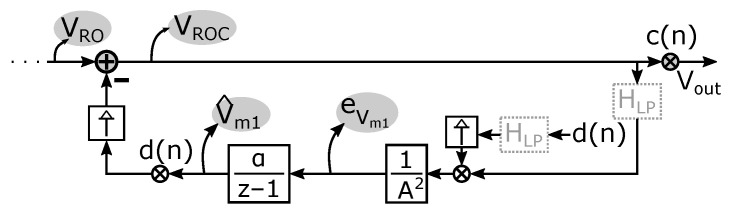
Detailed representation of one spread-spectrum offset reduction loop.

**Figure 9 sensors-22-06069-f009:**
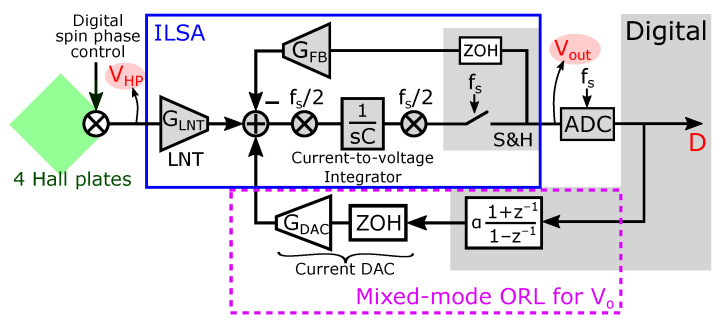
Simplified block diagram of what is implemented on the prototype chip: the In-the-Loop Sampling Amplifier (ILSA) (shown in the blue frame), which serves as the amplifier of the readout chain in this paper, a mixed-mode ORL for $V_{o}$ (shown in the purple square) and four Hall plates (shown in green) with digital spin phase control. Important signals are highlighted in red.

**Figure 10 sensors-22-06069-f010:**
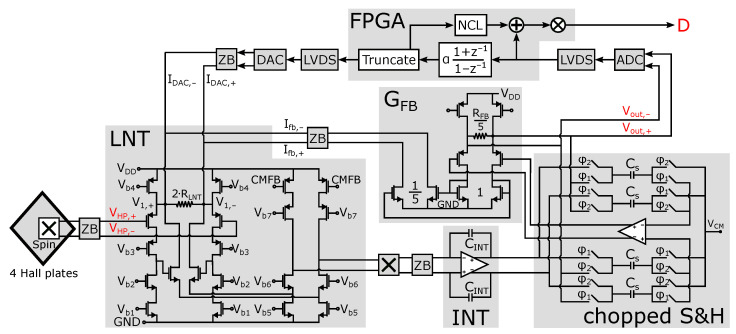
Implementation details of the readout chain used for experimental verification in Section 5 (adapted from (Figure 11 in [13])). Important signals are highlighted in red.

**Figure 11 sensors-22-06069-f011:**
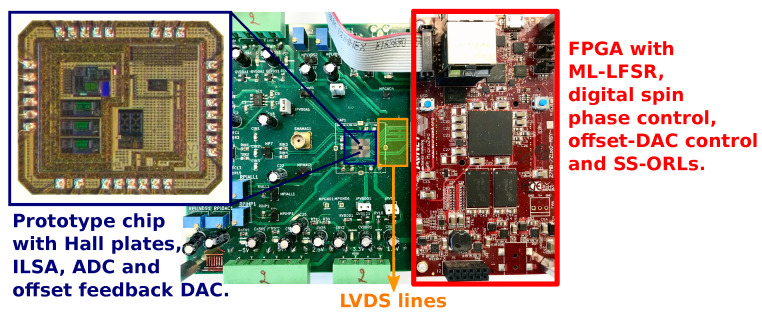
Photo of the test PCB containing the wire-bonded ILSA prototype chip [13] (shown zoomed-in in blue frame) and the FPGA (red board). In addition, the LVDS lines forming the communication between the chip and FPGA board are visible in the orange frame.

**Figure 12 sensors-22-06069-f012:**
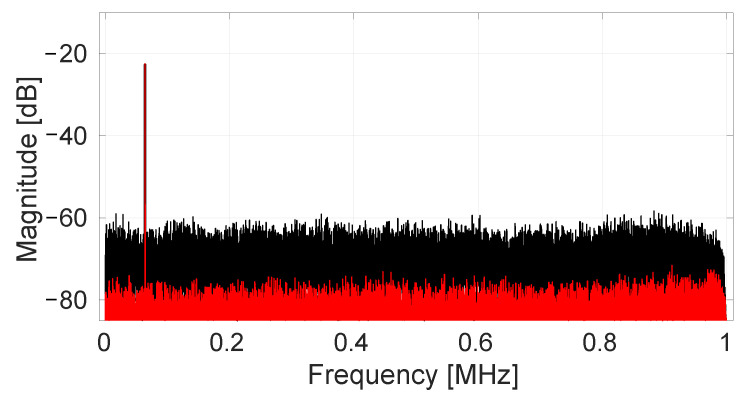
Measured output spectrum (32K FFT) for a magnetic input at 64 kHz with amplitude 780 μT. In black, no SS-ORL is active, and randomized offset is present modulated as pseudo noise. In red, the SS-ORLs are active, removing the randomized offsets. The 0 dB reference level corresponds to a full scale signal output amplitude of 0.8 V.

**Figure 13 sensors-22-06069-f013:**
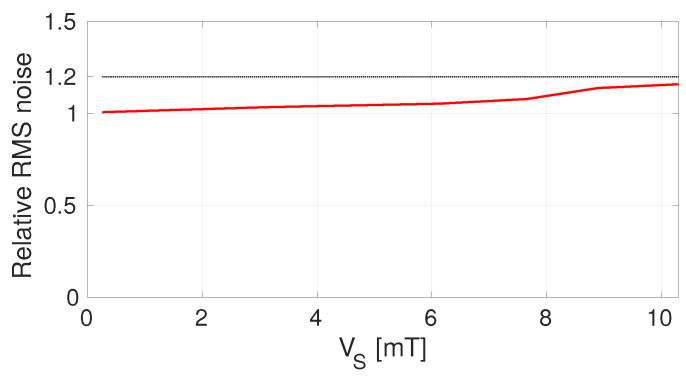
The measurement input-referred RMS noise divided by the measured intrinsic input-referred RMS noise level obtained by traditional spinning as a function of the DC input level (in red). The SS-ORLs from Figure 7 are active and settled, operating with an α value calculated using (Equation 23). In black, the 20%-threshold is shown.

**Figure 14 sensors-22-06069-f014:**
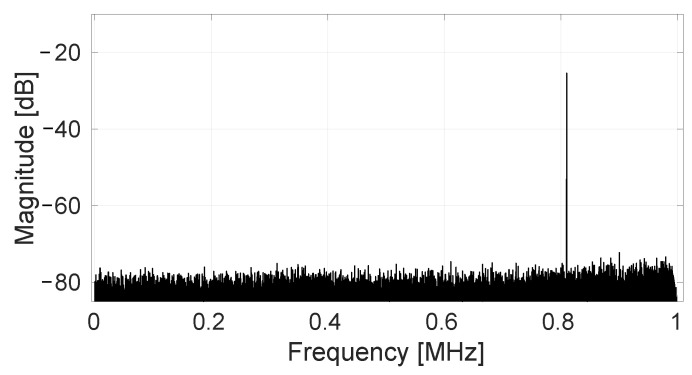
Measured output spectrum (32K FFT) for a magnetic input at 810 kHz with amplitude 780 μT. All SS-ORLs are active, removing the randomized offset. The 0 dB reference level corresponds to a full-scale signal output amplitude of 0.8 V.

**Figure 15 sensors-22-06069-f015:**
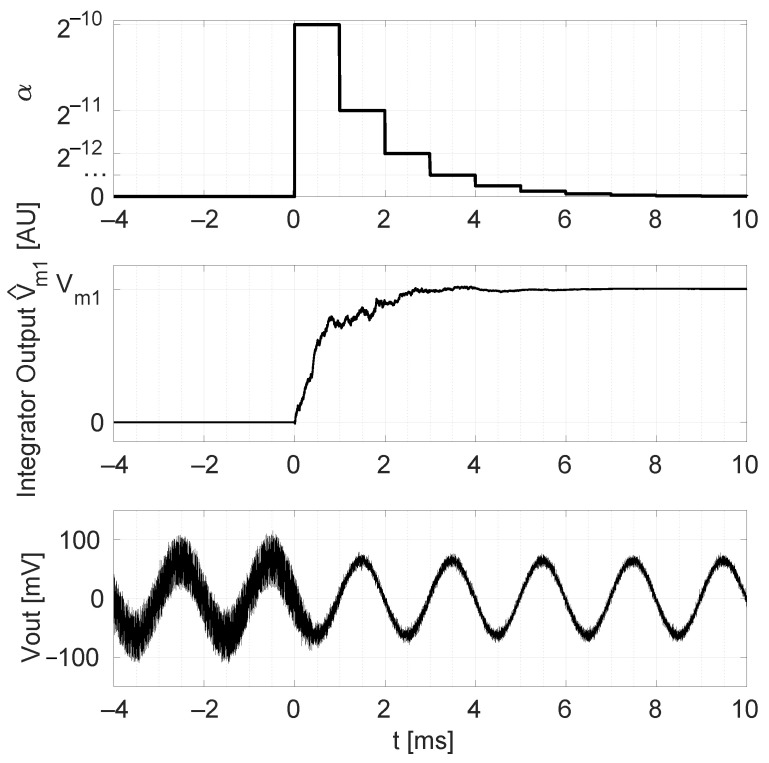
Time domain plots for the system with active SS-ORLs and stepwise changes in adaptation parameter: imposed time-evolution of the adaptation parameter α used in the SS-ORLs (**top**), the integrator output ${\hat{V}}_{m1}$ (**middle**), and $V_{\mathrm{out}}$ of Figure 7 (**bottom**). A magnetic field with amplitude 780 μT and frequency 500 Hz is applied.

**Figure 16 sensors-22-06069-f016:**
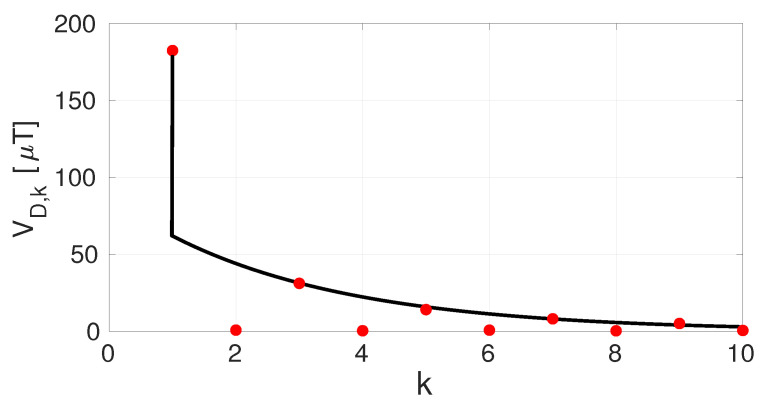
Mean μ of the measured $V_{D,k}$ offset terms as a function of k (red dots). An exponential fit to $V_{D,k}$, for k>1 and odd, is added.

**Figure 17 sensors-22-06069-f017:**
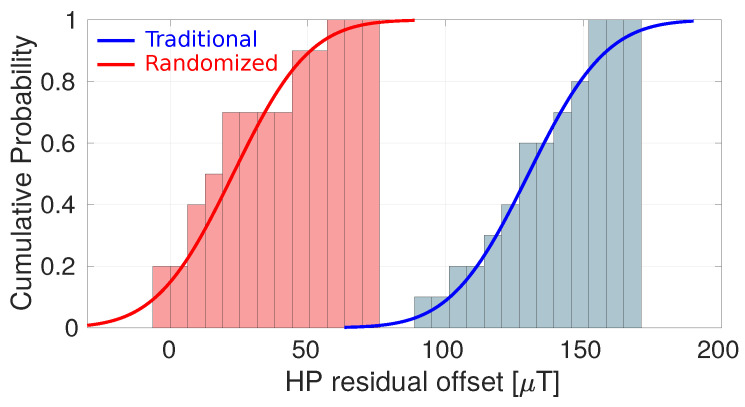
Cumulative distribution over 10 samples of the input-referred Hall plate residual offset with traditional four-phase spinning (blue) and randomized four-phase spinning (red). The full lines correspond to a cumulative Gaussian fitted to the associated cumulative distribution.

**Figure 18 sensors-22-06069-f018:**
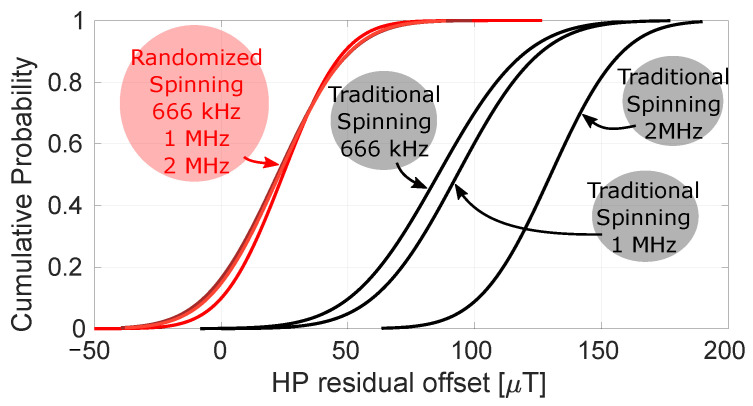
Fitted cumulative Gaussian distribution over 10 samples of the input-referred Hall plate residual offset with traditional four-phase spinning (black) and randomized four-phase spinning (red) for $f_{\mathrm{spin}}$ 666 kHz, 1 MHz and 2 MHz.

**Table 1 sensors-22-06069-t001:** Normal distribution fit on the input-referred offset terms from (Equation 10) in μT using 10 samples of the ILSA prototype chip with randomized Hall plate spinning.

	$V_{r}$	$V_{o}$	$V_{m1}$	$V_{m2}$	$V_{D,1}$	$V_{D,3}$	$V_{D,5}$	$V_{D,7}$
μ	23	37	122	116	183	31	14	8
σ	22	3455	40	51	11	2	1.3	0.8

**Table 2 sensors-22-06069-t002:** Comparison Table.

	This Work	[13]	[2]	[23]	[45]
	Hall	Hall	Hall	Hall	Hall
Inp. range [mT]	±10.6	±10.6	±12.5	±14.8	10
Noise [$\frac{\mathrm{nT}}{\sqrt{\mathrm{Hz}}}$]	52–63 *	55	430	136	-
Typ. offset [μT]	74 ** 23	68	40	>14,000	<50
BW [kHz]	820 ***	410	400	1000	7.8
Current [mA]	5.1 †	5.1	8	8.8	0.067
$T_{\mathrm{spin}}$ [μs]	0.5	1	25	0.0625	1

* Lowest noise for low input signals; ** input-referred offset is 74 μT if ILSA offset is included; the offset reduces to only 23 μT when ILSA offset is removed using a zero Hall plate bias current measurement; *** bandwidth after additional digital filtering (corresponding to the reported noise value); unfiltered 3 dB bandwidth is 960 kHz; † current consumption of prototype chip only.

## Data Availability

Not applicable.

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
