# Peer review of "A 2 MS/s Full Bandwidth Hall System with Low Offset Enabled by Randomized Spinning"

_sensors, 2022, doi:10.3390/s22166069_

Round 1

Reviewer 1 Report

(1). The entire analysis presented in Section 2.1 relies only on one citation to Ref [16]. Yet, the authors proceed to make the following claim on Page 5 (line 139) about Equation 9:  “This equation is valid, independent of the actual spinning scheme being used and shows that the residual offset Vr is indistinguishable from the magnetic signal VS. Hence Vr represents the fundamental offset suppression limit that can be reached by the spinning schemes of Fig. 1.” It is not clear to me on what logical or empirical basis the above claim is founded. Can the authors provide empirical evidence to justify this claim from data generated in this work?

 (2). In Section 3 where the authors have presented their approach to the Spread-spectrum Offset Reduction Loops, the way the Systems Derivation (Section 3.1) is present requires some modification. They write down Equation 12 and state that it can be derived within their stated conditions, without deriving it. They also indicate in the second term of Equation (12) that T{v0} is approximated to zero without giving any explanation howsoever of how this assumption would change their results if T{v0} had made a non-vanishing contribution to VRO.  I have a grave reservation with this style of research communication because it can lead to very misleading outcomes. Besides, this approach to the manipulation of analytical expressions makes it tedious for the reader to follow the discussion and the eventual conclusions. These thus make this work inaccessible to readers, in my view. I, therefore, suggest strongly that the authors should derive Equation 12 explicitly, and that they should include their derivation within the article itself either as an Appendix or as Supplementary material.

(3). Their entire analyses discussed herein are predicated on having what the authors have described as “good estimates” of the multiple independent offset parameters denoted by Vm1, Vm2, and VD,k. It is not clear to me what they mean by good estimates in the context of signal sensing. The authors should include a brief comment within the paragraph that discussed Equation (13) to indicate what constitutes such a good estimate, and how each of these is obtainable independently.

 (4). Another red flag is that the entire Section 3 has only cited one Reference. Worse still, the citation to Ref. [22] is irrelevant to the Systems Derivation presented in Section 3.1. It is instead more relevant to proposals for background self-calibration. The use of Ref. [22] as justification for their allusion to signal correlation is diversionary. They must rectify their Systems Derivation first and then argue their signal correlation argument correctly in a transparent presentation. This will make their work plausible.

 (5). Some of the Figure Captions:

(i). The caption of Fig 8 should be updated to show the meaning of the color-coded rectangular blocks explicitly.

(ii). The caption of Fig 10 is confusing. Multiple textual materials are displayed in blue, orange, and red-colored text, yet reference is made only to the “red board” in the caption.

Reviewer 2 Report

The authors report in this manuscript, with title “A 2MS/s Full Bandwidth Hall System With Low Offset Enabled by Randomized Spinning”, on the study on a Hall plate readout with a randomized four-phase spinning-current scheme. The goal of the work is to remove the maximum number of offset components, the offset associated with spike demodulation included. They want to innovate with this sensor by operating various offset-reduction loops in spread-spectrum mode, and the same time, allowing to remove error components without notching out any in-band signals. Finally, the authors performed a mixed-mode experimental circuit, with an analog part implemented in a custom integrated circuit and the digital control system is connected to the analog chip. Overall, the work is well researched and presented and deserves publication in the journal Sensors. Some minor points are:

1)       The introduction section is a bit long. It should be revised.

2)      Also in the introduction section the authors should indicate what CMOS means.

3)      In page 3, line 104, it is quoted the “Figure 1c”, but there are only Figures 1(a) and 1(b). Please, check this out. The same in line 118, there is no any “Figure 5.33” in the main text.

4)       In page 8, line 250, Figure 7 is quoted before Figure 6.

5)      The first time the alpha parameter (integration constant) appears in the main text is in Figure 7, but this parameter has not been defined before. Please, amend that.

6)      In page 17, line 463, the authors must explain with more detail why they took “the closest power of 2…”.

7)      The list of references should be increased.

Round 2

Reviewer 1 Report

The authors have satisfactorily addressed all the issues I raised in my initial report. I am therefore happy to recommend the publication of the revised manuscript in its current form.